# Real-time drone derived thermal imagery outperforms traditional survey methods for an arboreal forest mammal

**Ryan R. Witt**[1,2]*, **Chad T. Beranek**[1,2,3], **Lachlan G. Howell**[1,2], **Shelby A. Ryan**[1,2], **John Clulow**[1,2], **Neil R. Jordan**[4,5], **Bob Denholm**[3], **Adam Roff**[1,3]

1 School of Environmental and Life Sciences, University of Newcastle, Callaghan, New South Wales, Australia, 2 FAUNA Research Alliance, Kahibah, New South Wales, Australia, 3 Science Division, NSW Department of Planning, Industry and Environment, Newcastle, New South Wales, Australia, 4 Centre for Ecosystem Science, School of BEES, University of New South Wales (UNSW Sydney), Sydney, New South Wales, Australia, 5 Taronga Institute of Science and Learning, Taronga Conservation Society Australia, Taronga Western Plains Zoo, Dubbo, New South Wales, Australia

* ryan.witt@newcastle.edu.au

**Data Availability Statement:** All relevant data are within the manuscript and its Supporting information files.

## Abstract

Koalas (*Phascolarctos cinereus*) are cryptic and currently face regional extinction. The direct detection (physical sighting) of individuals is required to improve conservation management strategies. We provide a comparative assessment of three survey methods for the direct detection of koalas: systematic spotlighting (Spotlight), remotely piloted aircraft system thermal imaging (RPAS), and the refined diurnal radial search component of the spot assessment technique (SAT). Each survey method was repeated on the same morning with independent observers (03:00–12:00 hrs) for a total of 10 survey occasions at sites with fixed boundaries (28–76 ha) in Port Stephens (*n* = 6) and Gilead (*n* = 1) in New South Wales between May and July 2019. Koalas were directly detected on 22 occasions during 7 of 10 comparative surveys (Spotlight: *n* = 7; RPAS: *n* = 14; and SAT: *n* = 1), for a total of 12 unique individuals (Spotlight: *n* = 4; RPAS: *n* = 11; SAT: *n* = 1). In 3 of 10 comparative surveys no koalas were detected. Detection probability was 38.9 ± 20.03% for Spotlight, 83.3 ± 11.39% for RPAS and 4.2 ± 4.17% for SAT. Effective detectability per site was 1 ± 0.44 koalas per 6.75 ± 1.03 hrs for Spotlight (1 koala per 6.75 hrs), 2 ± 0.38 koalas per 4.35 ± 0.28 hrs for RPAS (1 koala per 2.18 hrs) and 0.14 ± 0.14 per 6.20 ± 0.93 hrs for SAT (1 koala per 43.39 hrs). RPAS thermal imaging technology appears to offer an efficient method to directly survey koalas comparative to Spotlight and SAT and has potential as a valuable conservation tool to inform on-ground management of declining koala populations.

## Introduction

The koala (*Phascolarctos cinereus)* is a vulnerable cryptic, and mostly nocturnal, arboreal marsupial that often occurs in low densities, and can be laborious and costly to detect in the wild [1–3]. The surviving koala populations endemic to New South Wales (NSW) have been

**Funding:** This work was funded by Remote Sensing and Landscape Science, Science Division, NSW Department of Planning, Industry and Environment and the University of Newcastle Centre for Creative Industries. Follow-up spotlighting line transect surveys used to calculate f(0) as per S1 were funded by Taronga Conservation Society, Australia. The funders had no role in study design, data collection and analysis, decision to publish, or preparation of the manuscript. The funders provided support in the form of salaries for authors [CTB, BD, NRJ, AR], but did not have any additional role in the study design, data collection and analysis, decision to publish, or preparation of the manuscript. The specific roles of these authors are articulated in the 'author contributions' section.

**Competing interests:** The authors have declared that no competing interests exist.

tabled for extinction by 2050 without urgent conservation and government intervention [4]. This follows sustained pressure caused by the European colonisation of Australia which has promoted koala population reductions of more than 50% in the past 232 years [1, 5], a factor that was most recently exacerbated by the loss of at least 5000 koalas in the 2019/2020 black summer bushfires [4, 6]. To optimise recovery effort and conservation management strategies, an efficient and effective means of surveying wild koala populations is urgently required.

In common with all species, the selection of survey method(s) is a critical factor influencing koala detectability [7], the accuracy and detail of the data, and the extent to which the results can be relied upon to inform conservation practice [1, 3]. Koala detection probability is influenced by occupancy, vegetation density, terrain accessibility, koala tree preferences, surveyor experience, and survey effort [8–10]. Survey methods typically vary in detection success relative to the degree of effort required, and the selection of specific survey approaches relies heavily on budget, which is an increasing concern given the funding landscape for Australian terrestrial vertebrate fauna conservation [11, 12]. Ideally, the selected survey method(s) for monitoring a threatened species in decline should ensure detection success relative to effort and cost for the most efficient outcome [13].

Methods for surveying for koalas include both indirect and direct techniques. Indirect techniques include faecal pellet or scat surveys [14–19], scat dog detection surveys [20], community/postal surveys [16, 21, 22], public participatory mapping [23] and acoustic monitoring [24]. These rely on evidence of occupancy and activity such as scratch marks, scats, citizen sightings, and bellows [3, 20, 22, 24, 25]. In contrast, the application of direct techniques, such as diurnal strip-width [10, 26, 27] and line-transects [10], nocturnal line-transects [3], and emerging remotely piloted aircraft system thermal imaging (RPAS, also known as drones) technologies [8, 28–30], require surveyors to physically sight koalas and this is preferable for density and population estimates [3].

Since koalas have a large home range and may be dispersed across the landscape in low densities [31], indirect survey techniques are often viewed as low-cost alternatives to direct techniques [2, 14]. From a management standpoint, there has been a calculated financial trade-off between systematic spotlighting and diurnal line-transects for accurate detection within localised habitat and the need to collect wide spatial landscape data [2, 32]. Thus, many koala monitoring programs have focused on faecal pellet surveys that collect presence/absence and activity data that is then used in occupancy modelling [32] to make inferences on a population [2, 17]. Faecal pellet surveys have been useful to identify koala feed tree preferences [33, 34] develop koala habitat maps [22, 25, 35–37], and, using conversion factors and a number of assumptions, have also been used to calculate precursory population density estimates [14, 17, 38–40].

Although indirect surveys have been useful, they are limited in that they do not provide the critical population demographics data needed to understand how imperilled koala populations have become (e.g. age, sex, recruitment, incidence of disease). A recent parliamentary inquiry into the status of koala populations and habitat in NSW has found that following the historical decline and the losses to koala populations in the recent bushfire season that the estimated number of 36,000 extant koalas in NSW is outdated and unreliable [4]. An understanding of population demographics will be required to develop the innovative conservation management strategies and interventions needed to secure the most imperilled koala populations in NSW. The limited budget available for conservation intervention, and the rate of continuing koala decline, means that it has become essential for conservation practitioners to develop new or optimised direct survey techniques for monitoring koalas.

Prior to the 2019/2020 bushfire season, an indirect survey method known as the spot assessment technique (SAT) that relies upon faecal pellets and chance sightings had been widely implemented to determine koala occupancy or activity patterns. To circumvent the financial challenges posed by direct survey methods, the SAT was refined to include a diurnal radial search as supplement (sometimes referred to as a koala point survey) for the direct detection of koalas [18] to calculate density [39, 40], but the method has not yet appeared in, and its validity has not yet been assessed in the scientific literature.

With technological advances, there are opportunities to move toward direct survey methods for monitoring koalas that have the potential to produce cost-effective, accurate, fine scale spatial landscape data. For example, RPAS can be coupled with sensors and infrared detectors that capture high-resolution thermal images [41]. RPAS are particularly promising for surveying wildlife [42, 43] and have been accurate and cost-effective in determining abundance [43] and for detecting a range of species including arboreal mammals such as monkeys [44] and koalas [8, 28–30].

RPAS thermal imaging technology has been used to initiate the development of automated machine learning algorithms that are designed to detect koalas from thermal imagery data [8, 28]. These algorithms were trained by surveying radio-tracked koalas in Petrie, Queensland and detections were validated by ground survey data collected on the same day [8]. Additionally, the method was employed to detect koalas during several one hectare plot surveys of a uniform moderately dense *Eucalyptus globulus* plantation on Kangaroo Island, South Australia [28]. In both cases thermal signatures were not validated in field, but rather during *ex situ* automated and manual frame by frame examination of footage post-hoc [8, 28].

In contrast, our recently published RPAS protocol offers real-time detection and validation of wild koalas by a combination of on-ground observation and the collection of 4K footage reviewed in the field [29]. We found that in the winter months a koala was on average 17.1 ± 2.7% brighter than the surrounding canopy vegetation [29]. Coincidently, our RPAS method has some parallels with the method described in Leigh *et al*. 2019 [30] which surveyed 20 ha of parkland vegetation (dense forest, open scrubland, grass fields) in south-east Queensland. In this study, which we were not aware of at the time of publishing Beranek *et al*. 2020 [29] or for the design of the present study, RPAS koala detections were ground truthed using in field observations from diurnal line-transects and were then combined with expert elicitation and public observation data to develop a statistical model that better predicts koala distribution [30].

RPAS thermal imaging technology has achieved koala detection in all seasons of the year in parkland and plantations, but has only achieved detection of wild koalas in natural bushland in winter [29], and this work was completed alongside the present study. RPAS thermal imaging technology for both the manual and automated detection of koalas is still in development, seasonal differences in precision and koala detection at variable speed and altitude have not been investigated [8, 28–30], and machine learning algorithms require further training [8, 28]. Despite this, and a claim of 'higher precision achieved' against traditional ground survey methods [8, 28], RPAS-derived methods have not been formally compared to other more widely used techniques.

We aimed to test our real-time in-field RPAS survey protocol described in Beranek *et al*. 2020 [29] to assess the effectiveness and efficiency of RPAS thermal imaging sensors compared to the refined SAT [18] and a nocturnal line-transect systematic spotlighting technique for detecting koalas. We also aimed to undertake a RPAS area census of each survey site and compare precursory density estimates and tree use information for each survey method.

## Materials and methods

### Animal ethics statement

The project was conducted under the NSW Office of Environment and Heritage Animal Ethics Committee license: AEC190312/06. Consent from private and government land managers were obtained.

### Site selection and timing

The comparative field surveys were conducted at six sites in Port Stephens, NSW, located ~30 km north of Newcastle (Fig 1), and at one site at Gilead, NSW, located ~10 km south of Campbelltown. Each study site was composed of a quadrat from 28–76 ha in size (Fig 2). Given that koalas in the survey area are thought to have an average home range size of 39.5 ha [31], we assumed that for all sites that individuals were present within the site area at the beginning of each survey would remain within the fixed boundaries for the entire survey period. There were three selection criteria for the sites of the experiment; recently reported activity of koalas, an abundance of primary feed trees, and accessibility. In Port Stephens, there was an additional criterion to select only sites considered either preferred or supplementary koala habitat as per Lunney *et al.* 1998 [22]. The vegetation at all sites was comprised of either open woodland or swamp forest containing primary koala feed trees, including swamp mahogany (*Eucalyptus robusta*) and/or forest red gum (*Eucalyptus tereticornis*).

At each site, a systematic spotlighting nocturnal line-transect (Spotlight) survey, RPAS survey, and SAT survey were conducted on the same morning in a fixed sequence (Fig 2). Spotlight surveys started at 03:00–04:00 hrs and finished no later than 06:00 hrs. RPAS surveys were conducted from 04:00–05:00 hrs and continued until the site was completed, no later than 07:30 hrs. The SAT surveys were conducted between 08:00–12:00 hrs. Each site was independently surveyed on either one occasion (Site PS-2, PS-3, PS-5, G-1), or on two occasions over two consecutive days (PS-1, PS-4, PS-6) to increase sample size and inform detection probability. Each survey method was conducted by randomised independent observers, except for RPAS surveys in which the pilot and boom operator was fixed. No communication was transferred between surveys teams during the study. All sites were surveyed in the winter

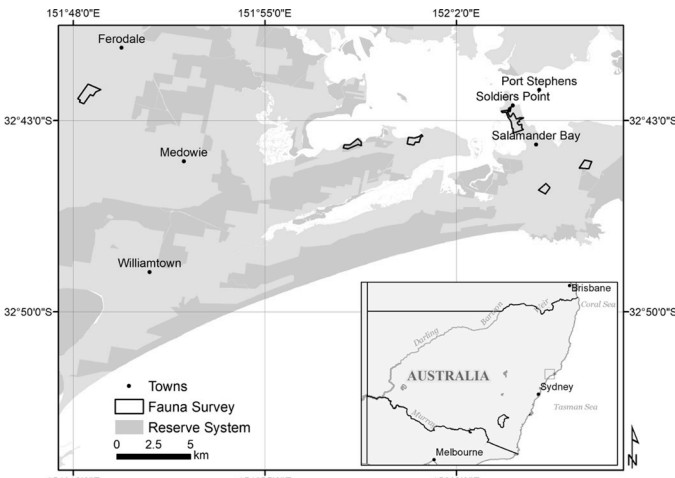

**Fig 1. Study site map and survey locations in Port Stephens on the east coast of New South Wales (NSW), Australia.**

**Fig 2. Conceptual diagram of the three survey methodologies (Spotlight, RPAS, SAT) conducted within a fixed site quadrat (28–76 ha in size) at each survey location on the same morning in Port Stephens and Gilead, NSW, Australia.** (a) Systematic spotlighting nocturnal line-transect (Spotlight) spaced ~100 m apart with an observation distance of 50 m perpendicular to the left and right of the transect line, 03:00–06:00 hrs; (b) remotely piloted aircraft system thermal imaging (RPAS) flown in a lawn mower pattern as per Beranek *et al.* 2020 [29], 04:00–07:30 hrs; (c) the refined spot assessment technique (SAT) a grid-based arrangement at a frequency of one SAT survey per 8.6 ± 1.21 ha, 08:00–12:00 hrs.

period May to July in 2019 in clear weather (ambient temperature: 11.08 ± 0.67 ˚C; wind speed maximum: <11 m/s).

## Systematic spotlight surveys

While line-transects to detect koalas have occurred diurnally [10], nocturnal line-transects have been shown to achieve higher detection rates [3]. We conducted systematic Spotlight surveys using head torches (Led Lenser, H14R) to scan vegetation for detection of reflected eye shine, similar to the method reported by Wilmott *et al.* 2019 [3]. Each transect line was walked concurrently by one or two observers as per Buckland *et al.* 1993 [45]. In Dique *et al.* 2003 [10] 90% of koala observations along line-transects were detected within 50 m of the transect line. Each transect line was therefore spaced ~100 m apart and had a theoretical maximum observation distance of 50 m perpendicular to the left and right of the transect line which ensured observers walking perpendicular transects were unlikely to report detection of the same animal (Fig 2). For each koala sighting, the occupied tree species was identified [using: 46, 47] and the GPS position were recorded (GARMIN eTrex) to avoid double counts and for comparison with RPAS survey thermal detection.

## RPAS surveys

At each site we deployed a quadcopter drone (DJI Matrice 200 v2). Flights were programmed using DJI Pilot (Android) and flown in a lawn mower pattern (parallel linear line-transects) with 10% overlap (Fig 2) as per Beranek *et al.* 2020 [29]. Thermal signatures were validated post-spotlighting by on-ground observers and by RPAS real-time visualisation of a suspected koala using a 4K colour camera as described in Beranek *et al.* 2020 [29].

## Spot assessment technique

We conducted SAT surveys in teams of two to four surveyors as per Phillips and Callaghan 2011 [19] which defines grid layout and centre tree selection. At each site, a grid-based arrangement was used at a frequency of one SAT survey per 8.6 ± 1.21 ha (Fig 2). To eliminate survey bias there was no repetition between a surveyor and a given SAT grid point or a surveyor and a known koala location determined by Spotlight or RPAS surveys. For each grid point the latitude and longitude were obtained and navigated to by a handheld GPS (GARMIN eTrex). In addition to the SAT, at each grid point we carried out a 25 m fixed radial search

from the centre tree for direct detection of koalas as prescribed as a SAT refinement in Phillips and Callaghan 2014 [18]. Each SAT grid point was ~224 m apart, a finer frequency than is typically used in grid-based SAT surveys [48], to maximise the potential to detect koalas and obtain occupancy and activity data.

## Density calculations

Density was calculated for each method per survey and compared to the calculated 'naïve density'.

Density ($D$) was calculated from line transects with the following equation: $D = \frac{n\hat{f}(0)}{2L}$, where $n$ is the number of koalas detected, $\hat{f}(0)$ is the 'sightability parameter' estimated as the probability of sightings at a distance of 0 m from the line, and $L$ is the length of the line transect sampled. A sample size of 60 to 80 sightings is needed to generate a statistically valid value for $\hat{f}(0)$ [10, 45, 49, 50]. As the present study did not attain the appropriate number of sightings to generate $\hat{f}(0)$, we calculated $\hat{f}(0)$ based on 96 koala observations collected across 73 follow-up spotlighting line-transect surveys (200 m in length) completed in Port Stephens between January and July 2020, $\hat{f}(0) = 0.043425$ (S1 File).

Density ($D$) was calculated from RPAS surveys with the following equation: $D = \frac{n}{A}$, where $n$ is the number of koalas detected and $A$ is the area covered by the RPAS.

Density ($D$) was calculated from the SAT surveys with the following equation, inferred from Phillips *et al.* 2007 [40]: $D = \frac{n}{S(\pi r^2)}$, where $n$ is equal to the number of koalas detected, $S$ is the number of SAT surveys (grid points) that occurred within the study site (~1 per 8.6 ha), and $r$ is equal to the length of the radial search conducted at each survey (25 m).

The 'naïve density' ($D_n$) of each site was calculated by the following equation: $D_n = \frac{n_u}{A_t}$, where $n_u$ is the number of unique koalas detected with all methods throughout a survey, and $A_t$ is the area surveyed by all methods, but does not include overlap.

## Effort efficiency calculations

A measure of effort for each survey method based on person time to detection was calculated. Effort was forecasted for each survey method based on records of paid staff/volunteer person minutes (actual survey effort) and mean on-site time (± 30 mins) for all personnel for all sites for the total survey period.

In addition, a measure of detectability relative to effort for each survey method was calculated. This gave an average measure of effort for successful detection and was used to determine the average total time required with each method to detect each koala. This was calculated as efficiency ($E$) by the following equation: $E = \frac{\bar{x}n_u}{\bar{x}e_{ph}}$, where $\bar{x} n_u$ is the mean number of unique koalas detected by the method per survey site, and $\bar{x}e_{ph}$ is the mean effort in person hours to complete the method per survey site. The economic implications and cost comparisons between the methods analysed here are subject to an ongoing investigation, Howell *et al.* (in prep).

## Statistical analysis

A general linear model with a Poisson distribution was used to determine differences in the number of koala detections per survey between each method ($n$ = 7 per survey). A quasi-Poisson distribution (dispersion parameter = 0.95) was used to test over-dispersion. Survey locations that did not have a direct koala detection across all methods were removed from the dataset ($n$ = 3 of 10 survey replicates) as the naïve density was assumed to be zero. Data from

all surveys was considered independent, including where repeat surveys occurred at a site on a separate evening and did not utilise the same observers.

An ANOVA using a likelihood-ratio chi-squared test statistic was used to determine an overall difference in koalas detected between each method. One-way ANOVA tests were used to determine the difference in probability of visual detection and total effort between each method. The Chi-squared test was used to assess the difference in presence/absence of koalas between each method. Density calculations were performed in Microsoft Excel (Version 16.42), general linear modelling was completed using RStudio (1.2.5033), ANOVA testing was completed using JMP (SAS Institute Inc.), and Chi-squared tests were completed in SPSS (IBM Corporation Software Group). All statistical testing was based on a significance level of $P < 0.05$; all reported p-values for chi-square tests were two-sided and based on exact tests. All reported mean-values were given as $\bar{x} \pm$ standard error of the mean (SEM).

## Results

### Direct visual koala observations

Koalas were directly observed on 22 occasions over seven of the 10 comparative surveys (Spotlight: $n = 7$; RPAS: $n = 14$; and SAT: $n = 1$), for a total of 12 unique individuals. In total, Spotlight detected four of the 12 individuals, while RPAS and SAT detected 11 and one respectively. Three of the surveys revealed an absence of koalas across all survey methods. Including detections from repeated surveys, RPAS detected the highest number of koalas (Fig 3). In the seven surveys koalas were detected by any method, RPAS had detections in all instances, while Spotlight had detections in four, and the SAT had detections in only one. Effective detectability (Fig 3) per site was $1 \pm 0.44$ koalas per $6.75 \pm 1.03$ hrs for Spotlight, $2 \pm 0.38$ koalas per $4.35 \pm 0.28$ hrs for RPAS and $0.14 \pm 0.14$ per $6.20 \pm 0.93$ hrs for SAT. There was an overall significant difference found in the number of koala detections between the three survey types ($\chi^2_{2,18} = 13.469$, $P = 0.001189$). There was a statistical difference in the number of detections between RPAS and SAT (Tukey HSD, Z ratio = 2.55, $P = 0.029$), but not between RPAS and Spotlight (Tukey HSD, Z ratio = 1.50, $P = 0.292$) or Spotlight and SAT (Tukey HSD, Z ratio = 1.82, $P = 0.163$).

### Density estimates

Density estimates (Table 1) were most closely associated to the naïve density in the RPAS method and were equal to the naïve density on 6 of 7 occasions (PS-2.1, PS-3.1, PS-4.2, PS-6.1, PS-6.2, G-1.1). The RPAS underestimated density at one site (PS-4.1) by a factor of 3.05. Both

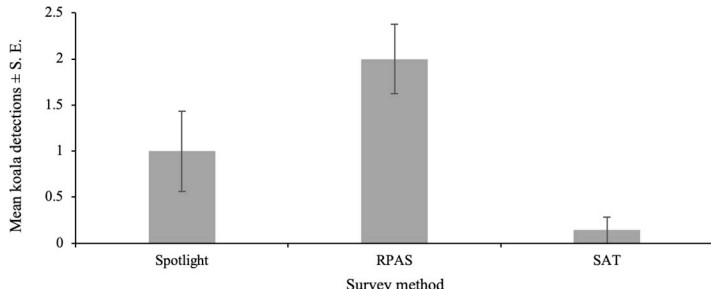

**Fig 3. Mean koala (*Phascolarctos cinereus*) detections per survey by method (Spotlight, RPAS, SAT) across low density peri-urban sites in Port Stephens ($n = 6$ per method) and Gilead ($n = 1$ per method) on the east coast of NSW, Australia.**

**Table 1. Koala (*Phascolarctos cinereus*) density estimates by survey method: Spotlight, RPAS, SAT per site.**

| Site (hectares) | Unique koala count | Spotlight $D = \frac{n\hat{f}(0)}{2L}$ | RPAS $D = \frac{n}{A}$ | SAT $D = \frac{n}{S(\pi r^2)}$ | Naïve density $D_n = \frac{n_u}{A_t}$ |
|---|---|---|---|---|---|
| *Port Stephens* | | | | | |
| PS-1.1 (76) | 0 | 0 | 0 | 0 | 0 |
| PS-1.2 (76) | 0 | 0 | 0 | 0 | 0 |
| PS-2.1 (57) | 1 | 0 | 0.018 | 0 | 0.018 |
| PS-3.1 (73) | 2 | 0 | 0.027 | 0 | 0.027 |
| PS-4.1 (47) | 3 | 0.168 | 0.021 | 0 | 0.064 |
| PS-4.2 (47) | 4 | 0 | 0.085 | 0.637 | 0.085 |
| PS-5.1 (48) | 0 | 0 | 0 | 0 | 0 |
| PS-6.1 (28) | 2 | 0.049 | 0.071 | 0 | 0.071 |
| PS-6.2 (28) | 2 | 0.049 | 0.071 | 0 | 0.071 |
| Mean koalas/ha | | 0.030 ± 0.019 | 0.033 ± 0.014 | | 0.037 ± 0.014 |
| *Gilead* | | | | | |
| G-1.1 (48) | 2 | 0.042[a] | 0.042 | 0 | 0.042 |

[a] $\hat{f}(0)$ was not used to calculate Spotlight density for Gilead in which the strip-width equation ($D = \frac{n}{2wL}$) was applied as per Dique *et al*. 2003 [10].

Values were calculated by equations given and described in the methods and are reported as koala density per hectare.

Spotlight and the SAT did not provide reliable estimates of density compared to naïve density. The SAT method provided a density estimate of 0 koalas in 6 of 7 surveys in which koalas were detected by the other methods. In the SAT survey (PS-4.2) one koala was detected in the radial search and this overestimated density by a factor of 7.49. The Spotlight method was equal to naïve density on one occasion (G-1.1) and differed from naïve density by a factor of 1.44 at two sites (PS-6.1, PS-6.2), a factor of 2.63 at one site (PS-4.1) and recorded a density value of 0 koalas in 3 of the 7 surveys that had koalas (PS-2.1, PS-3.1, PS-4.2). The mean naïve density for PS-1 to PS-6 was 0.037 ± 0.014 koalas/ha comparative to 0.030 ± 0.019 koalas/ha for Spotlight and 0.033 ± 0.014 koalas/ha for RPAS. The mean density for SAT could not be calculated.

### Probability of visual detection

When a koala was present within the survey site, the mean percent probability of detecting a koala was significantly different by survey method ($F_{2,17} = 8.6159$, $P = 0.003$). The probability of detection was 83.3 ± 11.39% when using a RPAS compared to 4.2 ± 4.17% for the SAT, which was significantly lower ($P = 0.002$). Whereas, the probability of detecting a koala for Spotlight was 38.9 ± 20.03% and similar in likelihood to both the RPAS method ($P = 0.083$) and SAT ($P = 0.198$).

### Effort efficiency estimates

For the survey period, the estimated mean effort per site was 6.75 ± 1.03 hrs for Spotlight, 4.35 ± 0.28 hrs for RPAS, and 6.20 ± 0.93 hrs for SAT. At sites that had a recorded presence of koalas, the total effort required to detect one koala (detectability ratio) was >6 hrs for Spotlight, >2 hrs for RPAS, and >43 hrs for SAT (Table 2).

### Site activity and occupancy analysis

We assessed 2014 trees for koala scats during the SAT surveys, 1774 in Port Stephens and 240 in Gilead (Table 3). At all sites in Port Stephens only 23 trees were active resulting in 13 of 61

**Table 2. The detectability success of the koala (*Phascolarctos cinereus*) based on effort requirements (person hours) of the three survey methods: Spotlight, RPAS, and SAT.**

| Survey method | Avg. Detections per method per site | Avg. person hours (effort) per method per site | Effort/success per method per site | Detectability ratio; 1 detection requires X effort (hours) |
|---|---|---|---|---|
| Spotlight | 1.00 | 6.75 ± 1.03 | 0.15 | 1 koala per 6.75 hrs |
| RPAS | 2.00 | 4.35 ± 0.28 | 0.46 | 1 koala per 2.18 hrs |
| SAT | 0.14 | 6.20 ± 0.93 | 0.02 | 1 koala per 43.39 hrs |

active grid points. At Gilead 6 trees were active resulting in 5 of 8 active grid points. In Port Stephens, SAT activity levels ranged from 3.3 to 13.3% ($\bar{x}$ 6.15 ± 0.91%), and in Gilead ranged from 3.3 to 6.7% ($\bar{x}$ 4 ± 0.67%).

The overall occupancy (presence/absence) was determined by both direct and indirect observations and confirmed at 4 of 10, 7 of 10, and 7 of 10 surveys for Spotlight, RPAS and the SAT respectively ($\chi^2_{2,30}$ = 2.500, *P* = 0.452). In all cases Spotlight and RPAS surveys confirmed occupancy by direct visual detection. The SAT surveys confirmed occupancy by indirect scat detection in all cases (Table 3), except for one survey (PS-4.2) which recorded scats and a single visual detection. All survey methods determined koala occupancy on three occasions (PS-4.1, PS-6.1, G-1.1) and failed to confirm occupancy on two occasions (PS-1.2, PS-5). On three occasions (PS-2.1, PS-3.1, PS-4.2) the RPAS and SAT determined koala occupancy and the Spotlight surveys failed to locate an animal. On one occasion (PS-6.2) the Spotlight and RPAS surveys directly detected koalas and the SAT failed to recover any evidence of koala presence. In contrast, on one occasion (PS1.1) Spotlight and RPAS surveys failed to directly detect a koala whereas the SAT survey determined low activity, 1 active tree in 3 separate grid points. In this instance, the validation of koala presence by the SAT method was due to the presence of old scats.

**Table 3. Koala (*Phascolarctos cinereus*) site and tree activity informed by the spot assessment technique (SAT) for each survey site in Port Stephens and Gilead, NSW, Australia.**

| Site (hectares) | Active/ total Grid Points (Score) | Active/total trees searched in all grids (Score) | Trees active (count) |
|---|---|---|---|
| *Port Stephens* | | | |
| PS-1.1 (76) | 3/8 (0.38) | 3/240 (0.01) | *Angophora costata* (1), *Eucalyptus siderophloia* (1), *Eucalyptus tereticornis* (1) |
| PS-1.2 (76) | 0/8 (0) | 0/239 (0) | |
| PS-2.1 (57) | 1/8 (0.13) | 2/241 (0.01) | *Eucalyptus pilularis* (1), *Eucalyptus robusta* (1) |
| PS-3.1 (73) | 2/4 (0.5) | 4/120 (0.03) | *Eucalyptus robusta* (4) |
| PS-4.1 (47) | 3/8 (0.38) | 6/240 (0.03) | *Angophora costata* (1), *Eucalyptus robusta* (4), *Eucalyptus tereticornis* (1) |
| PS-4.2 (47) | 3/8 (0.38) | 7/244 (0.03) | *Angophora costata* (1), *Eucalyptus pilularis* (3), *Eucalyptus robusta* (1), *Eucalyptus tereticornis* (2) |
| PS-5.1 (48) | 0/5 (0) | 0/150 (0) | |
| PS-6.1 (28) | 1/5 (0.2) | 1/150 (0.01) | *Eucalyptus robusta* (1) |
| PS-6.2 (28) | 0/5 (0) | 0/150 (0) | |
| Total: | 13/61 (0.21) | 23/1774 (0.01) | |
| *Gilead* | | | |
| G-1.1 (48) | 5/8 (0.63) | 6/240 (0.03) | *Eucalyptus crebra* (3), *Eucalyptus eugenoides* (1), *Eucalyptus punctata* (1), *Eucalyptus tereticornis* (1) |

Koala activity is reported as a probability score for the total number of active grids and active trees for each survey site.

### Tree utilisation in Port Stephens

In Port Stephens, the quantity of tree use observations (Table 4) was highest for the SAT which by indirect observation (scat searches) recorded 23 active trees. The SAT confirmed koala utilisation of five tree species including, 11 *Eucalyptus robusta*, five *Eucalyptus tereticornis*, four *Eucalyptus pilularis*, three *Angophora costata* and one *Eucalyptus siderophloia*. In contrast, the combined direct detection (Spotlight + RPAS) methods recorded 17 active trees and koala utilisation of 10 tree species, twice as many as the SAT. Koalas were found in five *E. robusta*, three *E. pilularis* and two *Melaleuca quinquenerva* and also occupied seven other species (Table 4). The direct methods determined koalas also utilised *M. quinquenerva, Casuarina glauca, Corymbia maculata, Eucalyptus parramattensis, Eucalyptus haemostoma* and *Livistona australis*.

*E. robusta* was found to be the most commonly used tree species in both direct and indirect methods. Tree utilisation probability (Table 4) for *E. robusta* was 0.40 of all observations, the four subsequent tree species utilised by koalas in Port Stephens were *E. pilularis* (0.18), *E. tereticornis* (0.13), *Angophora costata* (0.10) and *M. quinquenerva* (0.05).

Movement was observed in three koalas (*n* = 3 of 12) during the study, and these results were consolidated using the Spotlight and RPAS data. One koala was observed in an *E. robusta* at 03:00 hrs at 25 m height during a Spotlight survey (PS-4.1), and by 07:40 hrs was detected by the RPAS ~80 m from its original location ~35 m high in the crown of an *A. costata*. A second koala in the same survey (PS-4.1) was first observed in a *Livistona australis* at 04:40 hrs at 5 m height and subsequently moved twice before sunrise, first to a *E. tereticornis* at 06:00 hrs at 10 m height and then to a *C. glauca* at 07:00 hrs at ~1 m height. In a separate survey (PS-6.1) a koala was first observed in a *E. robusta* at 05:00 hrs at 8 m height, and by sunrise remained in the same tree but had moved into the crown to a height of 13 m. On the following evening (PS-6.1) the same koala had moved ~40 m and was first observed on the ground and selected a small *M. quinquenerva* at 05:20 hrs at 2 m height. Prior to sunrise the koala moved again ~20 m to a large *M. quinquenerva* and climbed to a height of 7 m and remained in that position after sunrise.

**Table 4. Comparison of tree species used by koalas (*Phascolarctos cinereus*) identified by direct observation (Spotlight, RPAS) or by indirect observation (SAT) within Port Stephens on the east coast of NSW, Australia (Site: PS-1 to PS-6).**

| Tree Species | No. of direct observations | No. of indirect observations | Total Observations | Utilisation Probability |
|---|---|---|---|---|
| | (Spotlight + RPAS) | (SAT) | | (PS-1 to PS-6) |
| *Eucalyptus robusta* | 5 | 11 | 16 | 0.40 |
| *Eucalyptus pilularis* | 3 | 4 | 7 | 0.18 |
| *Eucalyptus tereticornis* | 1 | 5 | 6 | 0.13 |
| *Angophora costata* | 1 | 3 | 4 | 0.10 |
| *Melaleuca quinquenerva* | 2 | 0 | 2 | 0.05 |
| *Casuarina glauca* | 1 | 0 | 1 | 0.03 |
| *Corymbia maculata* | 1 | 0 | 1 | 0.03 |
| *Eucalyptus parramattensis* | 1 | 0 | 1 | 0.03 |
| *Eucalyptus haemostoma* | 1 | 0 | 1 | 0.03 |
| *Eucalyptus siderophloia* | 0 | 1 | 1 | 0.03 |
| *Livistona australis* | 1 | 0 | 1 | 0.03 |
| Total trees active: | 17 | 23 | | |
| Total species active: | 10 | 5 | | |

## Discussion

We provide the first comparative assessment between systematic spotlighting (Spotlight), RPAS and the SAT for determining visual detection of individual koalas and estimating population density, as well as koala occupancy, and tree utilisation. We provide evidence that the RPAS method is optimal for detecting koalas compared to Spotlight and SAT when the spatial location and area are fixed, and the timing of all survey methods either overlap or are completed within a 9-hour period in winter.

Our results show that at 7 of 10 survey sites that koalas were present, koala detectability significantly differed by survey method. RPAS was found to significantly outperform the SAT, however, did not outperform Spotlight which did not statistically differ from either of the other methods. In terms of direct detection, our results validate that RPAS is an effective and efficient method for detecting koalas at low densities (1 koala per 2.18 hrs) and show that for similar investment in person hours a higher rate of direct koala detection can be achieved by Spotlight (1 koala per 6.75 hrs) compared to the SAT (1 koala per 43.39 hrs). We have also shown with repeat surveys at low density sites that RPAS was the optimal method for direct detection of individual koalas ($n$ = 11 of 12), compared to Spotlight ($n$ = 4 of 12) and the SAT ($n$ = 1 of 12).

The probability that RPAS detected all koalas known to occur within a site during the survey period was ~83%. This is higher than the manual detection rate of koalas by post-hoc analysis of RPAS-thermal imagery reported in Corcoran *et al.* 2019 [8] which was 52%, however differences in design prevent direct comparison. Unlike the present study, Corcoran *et al.* 2019 [8] surveyed an area of radio-tracked koalas and compared manual detection to automated detection using machine learning algorithms from RPAS derived imagery/footage to identify koala thermal signatures. In Hamilton *et al.* 2020 [28] which applied the RPAS method used in Corcoran *et al.* 2019 [8] an unknown number of potential koalas were detected in a moderately dense canopy where post-hoc automated machine learning analysis was able to achieve a 96% precision rate for the detection of koalas. In contrast to both studies, we used RPAS-derived imagery with in-field validation by on-ground ecologists during field surveys to detect an unknown number of potential koalas at each site. Due to this and the differences in habitat complexity between a plantation and a dense coastal vegetation system, it was not possible to calculate a level of precision for each of our survey methods for comparison with the post-hoc method utilised by Corcoran *et al.* 2019 [8] and Hamilton *et al.* 2020 [28].

The lower rate of detectability observed during Spotlight compared to RPAS surveys may be due to the complex and physically obstructive vegetation structure, observer error, the canopy density of *E. robusta*, or simply by the greater spatial coverage of the RPAS. For example, on the Tilligerry Peninsula in Port Stephens, *E. robusta* communities have previously affected spotlighting efficiency and detection success citing difficulty seeing into the dense canopy [51]. In the present study, we observed via a combination of direct and indirect methods that *E. robusta* was the tree species most likely to be utilised by koalas (40% of observations). This is well aligned with the literature. In Matthews *et al.* 2007 [52], koalas radio tracked in Port Stephens for up to 3 years were sighted in *E. robusta* on 26.7% of occasions and the preference to *E. robusta* is further supported by faecal pellet surveys of 3847 trees at 58 field sites [34].

We made several observations that may be important considerations regarding the detection probability of koalas for the design of future surveys. There was one instance where Spotlight resulted in a higher amount of detections compared to RPAS, and this occurred when there were high wind speeds recorded relative to all other survey occasions (>10.5 m/s). It is likely that the altitude of a koala within the tree relative to canopy cover could affect aerial

detection, and wind may affect RPAS detectability. We suggest that drone surveys do not exceed 10.5 m/s as per Beranek *et al.* 2020 [29].

Additionally, we observed koala movement (*n* = 3) occurring about half an hour before sunrise. It is known that koalas in Port Stephens and other regions tend to move to different trees during the evening compared to post-sunrise [31, 52, 53]. Koala movements have been found to occur from 20:00–21:00 hrs, 01:00–04:00 hrs and 05:00–06:00 hrs in Phillip Island, Victoria [54], and from 16:00–20:00 hrs and 02:00–04:00 hrs in south east Queensland [55]. However, no study in Port Stephens or other threatened koala populations in NSW has quantified hourly movement patterns. This data is likely to be useful for determining which time of day would result in the greatest detection probability of koalas with RPAS for each population, as we assume that koalas may be less detectable when moving, especially when moving between trees on the ground. To further refine RPAS surveys for koalas, it is important to investigate winter hourly movement patterns of koalas in each region so that RPAS surveys can be optimised and take into account any koala behaviour that could result in lower detectability.

In comparison to the other methods of direct detection, we were only able to directly detect one koala in 61 SAT radial searches. Even though the SAT was not found to be beneficial for detecting individual koalas, the SAT confirmed a low level of koala activity at all sites where koalas were directly detected, and one site where they were not directly detected (PS-1.1). Koala scats often persist for six months before decomposing, which enables a long period to obtain occupancy information [20]. It is our view that the SAT method remains optimal for determining site occupancy given the value in confirming transient koala habitat as shown at site PS-1.1 in which evidence of scats were recovered, but individual animals were absent at the site.

Koala tree use and preference evidence for Port Stephens has previously been determined using faecal pellet surveys [33, 34], and through radio tracking [52]. We found that for sites in Port Stephens (PS-1 to PS-6) that more trees species were identified as being used by koalas through the Spotlight and RPAS direct methods (*n* = 10) compared to the SAT via direct or indirect means (*n* = 5). This result is not surprising given that koala scat deposition is not constant and is known to be spatially and temporally disproportionate [14, 15] and only one koala was directly sighted using the SAT. Our results are supported by Ellis *et al.* 2013 [14] which found that tree use of radio tracked koalas in central and south east Queensland (*n* = 15) was only associated with scat presence on 49% of occasions, which could be increased to 77% if the tree was checked on the following day. Ellis *et al.* 2013 [14] further determined that on 23% of occasions trees utilised by koalas were not associated with a scat deposit. It is therefore likely that through direct survey methods, and particularly with improvement of RPAS surveys, that a greater understanding of koala tree selection could be achieved.

We found that direct methods (Spotlight; RPAS) are more accurate at estimating koala density than the refined SAT. RPAS surveys resulted in similar density estimates for six of seven survey sites, by comparison Spotlight surveys resulted in similar density estimates to the naïve density and the RPAS at only two sites. Although other studies have used strip and line-transects to estimate density [10, 26, 27], there have been no comparisons made between these methods and others in the primary literature. Our results show reasonable similarity between RPAS and Spotlight density estimates. We understand that for some practitioners, access to equipment might preclude the use of the RPAS method and suggest that the labour intensive Spotlight method when coupled with distance sampling (S1 File) is a useful alternative.

RPAS offer an opportunity to estimate koala density and populations sizes with reduced effort compared to other methods. This can be completed by either attempting an outright census of an area of interest, or by undertaking random sampling of a portion of the area and

using density estimates for extrapolation to the unsampled area. For example, a precursory population estimate of Port Stephens can be obtained from density estimates in this study. Given there is an area of 17145 ha of preferred and supplementary koala habitat [22], and the mean naïve density for PS-1 to PS-6 was 0.037 koalas/ha then the population of Port Stephens may be broadly estimated at ~634 koalas. However, these calculations are based on precursory density estimates from 329 ha of preferred koala habitat and do not consider koalas living in the urban landscape or reductions in habitat since Lunney *et al*. 1998 [22]. Greater sampling effort is therefore required to obtain confidence in the estimate, and to account for the detection error associated with the RPAS survey method.

The SAT did not accurately estimate koala density and reported either an absence of koalas on occasions in which there were koalas using the site ($n = 6$) or over-estimated koala density by a factor of 7.49 ($n = 1$). This may be due to several factors such as limited coverage of a sampled area, heterogeneity of forests that may confound estimates produced from extrapolation to unsampled areas, and a reduction in detection probability when attempting to detect animals in the canopy during diurnal searches [3].

Disparate to the results of our study, a consulting report [40] determined that koala density was similar between the SAT method (0.43 ± 0.06 koalas/ha) and a variant on the Dique *et al*. 2003 [10] diurnal strip/transect method (0.43 ± 0.11 koalas/ha). However, these calculations were based on an area of only 11.8 ha and were completed on a koala population that occurs in a higher density to the population we examined. Given that Spotlight was a more reliable predictor of density compared to the SAT for sites in Port Stephens and Gilead, we conclude that practitioners should refrain from using the SAT to estimate koala density.

## Conclusions

Conservation practitioners require new optimised technology to detect koalas at both a large spatial landscape scale and for assessment at the local scale to inform ongoing management of the declining koala populations on the east coast of Australia. The results of our study show that RPAS coupled with thermal imaging cameras are a promising efficient and effective alternative method to systematic spotlighting and the SAT for detecting koalas and estimating density at low density sites in the winter period. We highlight the potential application of RPAS to garner new insights into koala behaviour, movement, and tree utilisation preferences. Further, we have also shown that RPAS and systematic spotlighting are likely to be more accurate methods to estimate population density at low density sites than the SAT and suggest that the SAT should only be used to calculate precursory density estimates where funding other methods is not possible.

Future investment in the RPAS method should be considered highly beneficial for koala conservation. Unlike other methods of detection, the RPAS method has significant advantages and is likely to reduce the amount of time land managers are required to survey areas and allow ease in surveying difficult terrain, thereby providing a risk adverse method of ecological surveying. It is also likely that RPAS technology will be bolstered by advances in integrated machine learning algorithms with the sensor interface software that allows for automated koala detection [8, 28] but ideally this method would be optimised to allow for real-time in field detection in bushland across all seasons, and could also be coupled with other technologies for tracking released individuals such as koalas with VHF or GPS collars.

Once developed, standalone RPAS methods of automated detection of koalas may be useful as an efficient tool for presence/absence, habitat suitability/threat mapping, population counts and density measures, particularly across large spatial areas. Unless RPAS methods are combined with on-ground ecology [e.g. 29], automated machine learning technology alone may

never be useful to generate required knowledge for the fine-scale management intervention needed to ensure the longevity of declining koala populations (e.g. for understanding: recruitment, tree utilisation, rehabilitation survivorship, population demographics, incidence of disease). We expect that as RPAS technology improves it will become the optimal method for surveying koala populations and is likely to enable efficient strategic allocation of resources for koala conservation programs and for ongoing monitoring by public and private land managers.

## Supporting information

**S1 File. Sightability parameter $\hat{f}(0)$ calculation for estimating koala density.** The software program DISTANCE (release: 7.3) was used to calculate $\hat{f}(0)$ for Port Stephens from pooled distance data of 96 perpendicular observations of koalas collected across 73 line-transects (200 m in length) surveyed between January and July 2020, 6 repeats per site, on the Tomaree Peninsula.
(DOCX)

## Acknowledgments

We thank Paul Egglestone and Michael Cuneo (School of Creative Industries, University of Newcastle) for sharing practical RPAS knowledge. We thank field volunteers: Hayden Bond, Lachlan Burgess, Michael Day, Alex Callen, Daniel James, Paul Holmquest, Diane Kemp, Kate King, Lily Mickaill, Bridget Roberts, Claire Larkin, Samantha Sanders, Bonni Yare. We thank Carmel Northwood (Port Stephens Koalas) and Dorothea Willey (Tilligerry Habitat) for knowledge to inform site selection. We also thank NSW National Parks and Wildlife Services, Hunter Water, Port Stephens Council, Campbelltown City Council and Sydney Living Museums for site access.

## Author Contributions

**Conceptualization:** Chad T. Beranek, Adam Roff.

**Data curation:** Ryan R. Witt, Chad T. Beranek, Bob Denholm, Adam Roff.

**Formal analysis:** Ryan R. Witt, Chad T. Beranek, Lachlan G. Howell, Shelby A. Ryan.

**Funding acquisition:** Bob Denholm.

**Investigation:** Ryan R. Witt, Chad T. Beranek, Shelby A. Ryan, Bob Denholm, Adam Roff.

**Methodology:** Chad T. Beranek, Adam Roff.

**Project administration:** Chad T. Beranek, Adam Roff.

**Supervision:** Adam Roff.

**Validation:** Ryan R. Witt, Chad T. Beranek, Lachlan G. Howell, Shelby A. Ryan, Neil R. Jordan, Adam Roff.

**Visualization:** Ryan R. Witt, Chad T. Beranek.

**Writing – original draft:** Ryan R. Witt, Chad T. Beranek.

**Writing – review & editing:** Ryan R. Witt, Lachlan G. Howell, Shelby A. Ryan, John Clulow, Neil R. Jordan, Adam Roff.

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
