## [Decision Letter · Decision Letter 0]

22 Oct 2020

PONE-D-20-28042

Real-time drone derived thermal imagery outperforms traditional survey methods for an arboreal forest mammal

PLOS ONE

Dear Dr. Witt,

Thank you for submitting your manuscript to PLOS ONE. After careful consideration, we feel that it has merit but does not fully meet PLOS ONE’s publication criteria as it currently stands. Therefore, we invite you to submit a revised version of the manuscript that addresses the points raised during the review process.

The reviewers were very supportive of the manuscript, with only minor suggestions. If you can address these suggestions, the manuscript could be suitable for publication in PLoS ONE.

We look forward to receiving your revised manuscript.

Kind regards,

Mathew S. Crowther, Ph.D

Academic Editor

PLOS ONE

Journal Requirements:

2.Thank you for stating the following financial disclosure:

 [This work was funded by Remote Sensing and Landscape Science, Science Division, NSW Department of Planning, Industry and Environment and the University of Newcastle Centre for Creative Industries. Follow-up spotlighting line transect surveys used to calculate f(0) as per S1 were funded by Taronga Conservation Society, Australia. The funders had no role in study design, data collection and analysis, decision to publish, or preparation of the manuscript.].

We note that one or more of the authors is affiliated with the funding organization, indicating the funder may have had some role in the design, data collection, analysis or preparation of your manuscript for publication; in other words, the funder played an indirect role through the participation of the co-authors. If the funding organization did not play a role in the study design, data collection and analysis, decision to publish, or preparation of the manuscript and only provided financial support in the form of authors' salaries and/or research materials, please do the following:

Review your statements relating to the author contributions, and ensure you have specifically and accurately indicated the role(s) that these authors had in your study. These amendments should be made in the online form.

Confirm in your cover letter that you agree with the following statement, and we will change the online submission form on your behalf:

3.We note that [Figure(s) 1] in your submission contain [map/satellite] images which may be copyrighted. All PLOS content is published under the Creative Commons Attribution License (CC BY 4.0), which means that the manuscript, images, and Supporting Information files will be freely available online, and any third party is permitted to access, download, copy, distribute, and use these materials in any way, even commercially, with proper attribution. For these reasons, we cannot publish previously copyrighted maps or satellite images created using proprietary data, such as Google software (Google Maps, Street View, and Earth). For more information, see our copyright guidelines: http://journals.plos.org/plosone/s/licenses-and-copyright.

1.    You may seek permission from the original copyright holder of Figure(s) [1] to publish the content specifically under the CC BY 4.0 license. 

Additional Editor Comments (if provided):

The reviewers only have minor comments, that are relatively simple to address

Reviewers' comments:

Reviewer's Responses to Questions

**Comments to the Author**

1. Is the manuscript technically sound, and do the data support the conclusions?

Reviewer #1: Yes

Reviewer #2: Yes

2. Has the statistical analysis been performed appropriately and rigorously? 

Reviewer #1: Yes

Reviewer #2: Yes

3. Have the authors made all data underlying the findings in their manuscript fully available?

Reviewer #1: Yes

Reviewer #2: Yes

4. Is the manuscript presented in an intelligible fashion and written in standard English?

Reviewer #1: Yes

Reviewer #2: Yes

5. Review Comments to the Author

Reviewer #1: Comments to Author

The authors aims to assess the three survey methods for the direct detection of koalas: systematic spotlighting (Spotlight), remotely piloted aircraft system thermal imaging (RPAS), and the refined diurnal radial search component of the spot assessment technique (SAT). Generally, the authors found that RPAS system was the most effective at detecting individual koalas.

Although the general framework and the write up of the manuscript are good, I found several points to be addressed by the authors.

Introduction

The authors need to provide some justification why the previous methods are not affective. For example, they do not provide any information on the drawbacks of the indirect methods. This will also further cement their argument of using more sophisticated and costly equipment for animal surveys.

There is also a lack of information biological and ecological information on koalas. The audience should be made aware that the species is nocturnal and moves mostly at night. People in Australia may know this, but the rest of the world might not.

Methods

It would be good for the reader if there was some description of the habitat at each location (i.e. vegetation, climate).

Was temperature recorded during the survey? It can affect elevation of koala in the tree, as well as the infrared sensor.

Line 126: Very confusing writing 04:00 – 05:00 hrs. Why not 4 am to 5 am.

Line 133: Would seasonal variation in weather affect the effectiveness of each of the methods. There is no mention of that in the manuscript.

Line 202: Include the RStudio Team reference

I am confused why the results of the tree use were not actually included in the results, even though you talked about them in the discussion. It would be easier for the reader to have them in the results.

Discussion

Line 252: This needs to be brought up in the introduction

Line 253: This contradicts your results where there was significance between methods.

Line 295: Are you referring to studies at Port Stevens or in general. There are several studies that have reported on nocturnal movements of koalas. See Marsh et al. 2013

Is there any drawback on the use of RPAS system? (i.e. ambient temperature) and if so, it can be included in the discussion.

I really like the graphical overview of the methods in Figure 2.

Reviewer #2: Review: Real-time drone derived thermal imagery outperforms traditional survey methods for an arboreal forest mammal

Abstract:

L16: “difficult to monitor” – this is a broad statement. By cryptic, one is aware of the difficulties of detection. Perhaps remove this reference to monitoring, because the difficulties relate to a range of issues, many of which are not ecological (e.g. community perception, ethical clearance etc). Just a suggestion.

Introduction:

The introduction is concise and clear and was a pleasure to read.

Materials and methods

I presume there is a more formal description of the flight path than “lawn mower pattern”; if possible…?

Line 164: Density calculations, can you define the 2L in the text please?

Line 191: perhaps write as: Howell et al. (in prep)

Not being familiar with the technical details of the RPAS, these might be supplied in the supplementary data. Technical settings and specifications for detection using the RPAS (flight altitude, details of infrared settings and camera specifications to determine field of view) would be of great use to other researchers.

Results:

Line 212: It is understood that one cannot observe less than a complete koala, hence the effective detectability of two methods are rounded to whole units, but this is impossible for the SAT-based approach, so perhaps it would be simpler to standardise the effective detectability, perhaps to a unit of time?

The results don’t refer to the tree species, which was an interesting element found in the supplementary data. Given that some may not read the supplementary information, perhaps drawing attention to this interesting result is appropriate, in the main text.

Discussion

Line 280: Perhaps reword: as written it appears that the vegetation community cited a difficulty (again, this is merely a suggesting).

Line 292: It seems unlikely that a drone could fly adequately in conditions that were so severe as to drive a koala to seek lower branches. Just my observation – not supported by any facts here, but it seems implausible, given the conditions koalas routinely endure with little apparent distress. Unless you do have observations that support this contention (regarding wind speed), it may be more appropriate to state that the altitude of a koala within the tree, relative to canopy cover, could affect aerial detection, and wind may affect the drone (if you concur).

Line 296: The selection of day and night trees by koalas is also supported by the work of Melzer (1994 I think?) validating your conclusion here. A comment as to the relationship between koala activity and the likelihood of visual detection (e.g. spotlighting) is appropriate.

Line 320: The similar estimation of density resulting from RPAS and Spotlighting should be noted here. Although the effort is dissimilar, access to equipment may preclude the use of RPAS, so the information in Table 1 is useful and could be addressed here.

Line 348: your data suggest that the SAT is not a good choice for this task, despite its appropriateness for detecting occupancy, so you could be firmer.

Conclusions

Line 355: The reference to winter is perhaps important? This is the first mention of it, but I presume this has to do with thermal target image and contrast. If so, comments regarding this are more appropriate than (for example) comments regarding wind speed, in my opinion. If this is a key limitation of the technique, please provide some background.

Line 361: I am unsure, given your results, why you would conclude that the method you are using is not a far superior approach, particularly at the landscape level where cost – effectiveness is vital. The required improvements you list to do not appear to have limited your study, so I find this confusing. This is particularly the case with comments regarding imagery. As a result, your conclusions sow a seed of doubt in the reader that (in my case anyway) did not previously exist.

6. PLOS authors have the option to publish the peer review history of their article (what does this mean?). If published, this will include your full peer review and any attached files.

Reviewer #1: No

Reviewer #2: No

---

## [Author Response · Author response to Decision Letter 0]

27 Oct 2020

Dear Assoc. Prof. Crowther, 

Please find detailed herein responses to the reviewer comments for Manuscript No.: PONE-D-20-28042, ‘Real-time drone derived thermal imagery outperforms traditional survey methods for an arboreal forest mammal’. 

We thank you for providing suggested changes based on the Journal requirements. We also thank the reviewers for reviewing our paper. We have addressed your comments and each of the reviewer’s specific comments below and these can also be found in the track changed manuscript. 

Academic Editor: 

Author Response: We have refined the manuscript to fit with PLOS ONE’s style requirements. 

2. Thank you for stating the following financial disclosure: [This work was funded by Remote Sensing and Landscape Science, Science Division, NSW Department of Planning, Industry and Environment and the University of Newcastle Centre for Creative Industries. Follow-up spotlighting line transect surveys used to calculate f(0) as per S1 were funded by Taronga Conservation Society, Australia. The funders had no role in study design, data collection and analysis, decision to publish, or preparation of the manuscript.]. We note that one or more of the authors is affiliated with the funding organization, indicating the funder may have had some role in the design, data collection, analysis or preparation of your manuscript for publication; in other words, the funder played an indirect role through the participation of the co-authors. If the funding organization did not play a role in the study design, data collection and analysis, decision to publish, or preparation of the manuscript and only provided financial support in the form of authors' salaries and/or research materials, please do the following:

• Review your statements relating to the author contributions and ensure you have specifically and accurately indicated the role(s) that these authors had in your study. These amendments should be made in the online form.

• Confirm in your cover letter that you agree with the following statement, and we will change the online submission form on your behalf: “The funder provided support in the form of salaries for authors [insert relevant initials], but did not have any additional role in the study design, data collection and analysis, decision to publish, or preparation of the manuscript. The specific roles of these authors are articulated in the ‘author contributions’ section.”

Author Response: We have reviewed the statements relating to author contributions, and these are accurate. We confirm at that the funders provided support in the form of salaries for authors [CTB, BD, NRJ, AR], but did not have any additional role in the study design, data collection and analysis, decision to publish, or preparation of the manuscript. The specific roles of these authors are articulated in the ‘author contributions’ section.

3. We note that [Figure(s) 1] in your submission contain [map/satellite] images which may be copyrighted. All PLOS content is published under the Creative Commons Attribution License (CC BY 4.0), which means that the manuscript, images, and Supporting Information files will be freely available online, and any third party is permitted to access, download, copy, distribute, and use these materials in any way, even commercially, with proper attribution. For these reasons, we cannot publish previously copyrighted maps or satellite images created using proprietary data, such as Google software (Google Maps, Street View, and Earth). For more information, see our copyright guidelines: http://journals.plos.org/plosone/s/licenses-and-copyright.We require you to either (1) present written permission from the copyright holder to publish these figures specifically under the CC BY 4.0 license, or (2) remove the figures from your submission.

Author Response: Figure 1 is creative commons. The source is: Source: “NPWS Estate” and “Estuaries” by State Government of NSW and Department of Planning, Industry and Environment 2019 is licensed under CC BY 4.0. Made with Natural Earth.

4. Please include captions for your Supporting Information files at the end of your manuscript, and update any in-text citations to match accordingly. Please see our Supporting Information guidelines for more information: https://protect-au.mimecast.com/s/0mYWCYW8ZKUAGzzYfK21x0?domain=journals.plos.org.

Author Response: We have made changes to the supporting information. On recommendation from both reviewers the tree use and habitat results supplied as S2 and S3 is now included in the manuscript. We have retained S1 – File. We have adopted PLOS ONE’s style requirements and have included a caption at the end of the manuscript as requested. 

Reviewer 1: 

1. The authors aims to assess the three survey methods for the direct detection of koalas: systematic spotlighting (Spotlight), remotely piloted aircraft system thermal imaging (RPAS), and the refined diurnal radial search component of the spot assessment technique (SAT). Generally, the authors found that RPAS system was the most effective at detecting individual koalas.Although the general framework and the write up of the manuscript are good, I found several points to be addressed by the authors.

Author Response: We thank the reviewer for supporting our manuscript and providing suggestions for improvement in clarity. 

Introduction:

2. The authors need to provide some justification why the previous methods are not affective. For example, they do not provide any information on the drawbacks of the indirect methods. This will also further cement their argument of using more sophisticated and costly equipment for animal surveys.

Author Response: The authors agree, and thank the reviewer for the suggestion. We have added a paragraph that addresses the key drawback in the context on the current needs for koala recovery in Australia. See Paragraph L89-99. We have also edited pargraph L100-106 for flow. 

3. There is also a lack of information biological and ecological information on koalas. The audience should be made aware that the species is nocturnal and moves mostly at night. People in Australia may know this, but the rest of the world might not.

Author Response: We agree and have addressed this in the first sentence, L51-52, by adding nocturnal which implies that koalas are active and move at night. Now reads: ‘The koala (Phascolarctos cinereus) is a vulnerable cryptic, and mostly nocturnal, arboreal marsupial that often occurs in low densities, and can be laborious and costly to detect in the wild [1-3]’. This is the most pertinent aspect of koala biology that we previously excluded, and in the interests of a succinct paper we have not added other less relevant aspects of koala biology that are covered extensively in the existing literature. 

Materials and methods:

4. It would be good for the reader if there was some description of the habitat at each location (i.e. vegetation, climate).

Author Response: We have now added in the dominant vegetation structure for the sites. We have also added in ambient temperature information, and the weather variables that were used to standardise each survey. See L173-175: ‘The vegetation at all sites was comprised of either open woodland or swamp forest containing primary koala feed trees, including swamp mahogany (Eucalyptus robusta) and/or forest red gum (Eucalyptus tereticornis)’ and L184-186: ‘All sites were surveyed in the winter period May to July in 2019 in clear weather (ambient temperature: 11.08 ± 0.67 °C; wind speed maximum: <11 m/s)’.

5. Was temperature recorded during the survey? It can affect elevation of koala in the tree, as well as the infrared sensor.

Author Response: We recorded the ambient temperature during each survey. We have now added this information and we have ensured that we have mentioned that the surveys were limited to winter. See L184-186: ‘All sites were surveyed in the winter period May to July in 2019 in clear weather (ambient temperature: 11.08 ± 0.67 °C; wind speed maximum: <11 m/s)’. Despite this RPAS surveys for koalas have been achieved in all months of the year. See paragraph L134-141. 

6. Line 126: Very confusing writing 04:00 – 05:00 hrs. Why not 4 am to 5 am.

Author Response: We disagree with the reviewer; we have used standardised military time to remove any possible ambiguity. 

7. I really like the graphical overview of the methods in Figure 2.

Author Response: We thank the reviewer for the kind comment.

8. Line 133: Would seasonal variation in weather affect the effectiveness of each of the methods. There is no mention of that in the manuscript.

Author Response: It is possible that weather could have an impact on RPAS surveys, and to manage this we standardised our surveys by assessing methods on the same morning, and in the winter period. We only conducted surveys in clear weather to account for weather variables that limit the RPAS. This information is well covered in our paper Beranek et al. 2020 (ref: 29). Ambient temperature and wind speed has also been added to the methods. See L184-186: ‘All sites were surveyed in the winter period May to July in 2019 in clear weather (ambient temperature: 11.08 ± 0.67 °C; wind speed maximum: <11 m/s)’. We have now also added a sentence into the introduction to address the success of RPAS technology in detecting koalas in all seasons (See paragraph L134-141). However, the evaluation of methods by season has not formally been reported in the literature, and would require a further experiment. 

9. Line 202: Include the RStudio Team reference.

Author Response: It is not appropriate to add in RStudio here as we did not use this program for this purpose. This calculations were performed in excel, and we have added this information to the statistical analysis section. R Studio is referenced for an alternate purpose on L300 ‘general linear modelling was completed using RStudio (1.2.5033)’, 

Results:

10. I am confused why the results of the tree use were not actually included in the results, even though you talked about them in the discussion. It would be easier for the reader to have them in the results.

Author Response: We thank the reviewer for this suggestion. We have added in the tree use information at the end of the results section. See Sections from L358-406, and Table 3 and Table 4, and additions to the statistical analysis, L297-306.

Discussion:

11. Line 252: This needs to be brought up in the introduction. 

Author Response: We have moved to the introduction as suggested. See L62-64

12. Line 253: This contradicts your results where there was significance between methods.

Author Response: We thank the reviewer for drawing our attention to this error. We have amended the sentence to remove this contradiction. See L426-429: ‘Our results show that at 7 of 10 survey sites that koalas were present, koala detectability significantly differed by survey method. RPAS was found to significantly outperform the SAT, however did not outperform Spotlight which did not statistically differ from either of the other methods’.

13. Line 295: Are you referring to studies at Port Stevens or in general. There are several studies that have reported on nocturnal movements of koalas. See Marsh et al. 2013. 

Author Response: We have provided further information for clarity and have referenced two other studies including Marsh et al. (ref: 54, 55) that report on koala movements in other regions.

54. Marsh KJ, Moore BD, Wallis IR, Foley WJ. Continuous monitoring of feeding by koalas highlights diurnal differences in tree preferences. Wildlife Research. 2014;40(8):639-46.

55. Ellis WA, FitzGibbon SI, Barth BJ, Niehaus AC, David GK, Taylor BD, et al. Daylight saving time can decrease the frequency of wildlife–vehicle collisions. Biology letters. 2016;12(11):20160632.

14. Is there any drawback on the use of RPAS system? (i.e. ambient temperature) and if so, it can be included in the discussion.

Author Response: Whilst we agree with the reviewer that this is of interest to practitioners, there is no comparative drawback that would add value to a discussion that compares RPAS to Spotlight and SAT. We have already published our RPAS methodology for koala detection, see Beranek et al. 2020 (ref: 29). This paper suggests the technological limitations of the method, but all of these were accounted for in the design of our study. Commenting on how RPAS technology could be improved is outside of the scope of this paper, and has now been removed from the discussion on the recommendation of Reviewer 2.

Reviewer 2: 

Abstract: 

1. L16: “difficult to monitor” – this is a broad statement. By cryptic, one is aware of the difficulties of detection. Perhaps remove this reference to monitoring, because the difficulties relate to a range of issues, many of which are not ecological (e.g. community perception, ethical clearance etc). Just a suggestion.

Author Response: The authors agree. We have removed from “difficult to monitor” from the sentence. See L27: ‘Koalas (Phascolarctos cinereus) are cryptic and currently face regional extinction’.

Introduction: 

2. The introduction is concise and clear and was a pleasure to read.

Author Response: We thank the reviewer for the kind assessment of our introduction.

Materials and methods: 

3. Not being familiar with the technical details of the RPAS, these might be supplied in the supplementary data. Technical settings and specifications for detection using the RPAS (flight altitude, details of infrared settings and camera specifications to determine field of view) would be of great use to other researchers.

Author Response: We appreciate this suggestion by the reviewer, and whilst we agree, we have already published all of the suggested settings and specifications in Beranek et al. 2020 (ref: 29), which is available in Australian Mammalogy. We have stated that these specifications are available in an accessible location. See L229-233: ‘At each site we deployed a quadcopter drone (DJI Matrice 200 v2). Flights were programmed using DJI Pilot (Android) and flown in a lawn mower pattern (parallel linear line-transects) with 10% overlap (Fig 2) as per Beranek, Roff (29). Thermal signatures were validated post-spotlighting by on-ground observers and by RPAS real-time visualisation of a suspected koala using a 4K colour camera as described in Beranek, Roff (29).’ 

4. I presume there is a more formal description of the flight path than “lawn mower pattern”; if possible…?

Author Response: Lawn mower pattern is the terminology used in our earlier publication, Beranek et al. 2020, and thus have left for consistency in the language between our studies. We have added also added some content to improve clarity which includes a formal explanation. See L229-231: ‘Flights were programmed using DJI Pilot (Android) and flown in a lawn mower pattern (parallel linear line-transects) with 10% overlap (Fig 2) as per Beranek, Roff (29)’.

5. Line 164: Density calculations, can you define the 2L in the text please?

Author Response: We have defined L in text. See L261: ‘and L is the length of the line transect sampled’. The number 2 is a multiplier. 

6. Line 191: perhaps write as: Howell et al. (in prep)

Author Response: We have amended as suggested by the reviewer. See L286.

Results: 

7. Line 212: It is understood that one cannot observe less than a complete koala, hence the effective detectability of two methods are rounded to whole units, but this is impossible for the SAT-based approach, so perhaps it would be simpler to standardise the effective detectability, perhaps to a unit of time?

Author Response: Is essential to leave the calculation as less than 1 koala as this highlights the distinct inefficiency of this method, which is a key result. We have also already standardised to time here and in our efficiency calculations, See Table 2 and L38-40 in the abstract: ‘Effective detectability per site was 1 ± 0.44 koalas per 6.75 ± 1.03 h for Spotlight (1 koala per 6.75 hrs), 2 ± 0.38 koalas per 4.35 ± 0.28 hrs for RPAS (1 koala per 2.18 hrs) and 0.14 ± 0.14 per 6.20 ± 0.93 hrs for SAT (1 koala per 43.39 hrs)’.

8. The results don’t refer to the tree species, which was an interesting element found in the supplementary data. Given that some may not read the supplementary information, perhaps drawing attention to this interesting result is appropriate, in the main text.

Author Response: We thank the reviewer for this suggestion. We have added in the tree use information at the end of the results section. See Sections from L358-406, and Table 3 and Table 4, and additions to the statistical analysis, L297-306.

Discussion: 

9. Line 280: Perhaps reword: as written it appears that the vegetation community cited a difficulty (again, this is merely a suggesting).

Author Response: The authors agree with the reviewer. We have amended to improve the sentence. See L470: ‘The lower rate of detectability observed during Spotlight compared to RPAS surveys may be due to the complex and physically obstructive vegetation structure, observer error, the canopy density of E. robusta, or simply by the greater spatial coverage of the RPAS.’

10. Line 292: It seems unlikely that a drone could fly adequately in conditions that were so severe as to drive a koala to seek lower branches. Just my observation – not supported by any facts here, but it seems implausible, given the conditions koalas routinely endure with little apparent distress. Unless you do have observations that support this contention (regarding wind speed), it may be more appropriate to state that the altitude of a koala within the tree, relative to canopy cover, could affect aerial detection, and wind may affect the drone (if you concur).

Author Response: We concur with the reviewer, we have made changes to the paragraph in line with the reviewers comment. See Paragraph L479-484.

11. Line 296: The selection of day and night trees by koalas is also supported by the work of Melzer (1994 I think?) validating your conclusion here. A comment as to the relationship between koala activity and the likelihood of visual detection (e.g. spotlighting) is appropriate.

Author Response: We have added a reference for other regions at the end of the sentence so that the readers understand that this behaviour is not limited to Port Stephens. Melzer 1994 could not be found, perhaps it is a PhD chapter. Instead we have added a Journal article by the same author from 2011 (ref: 53), see L491-493: ‘Additionally, we observed koala movement (n = 3) occurring about half an hour before sunrise. It is known that koalas in Port Stephens and other regions tend to move to different trees during the evening compared to post-sunrise [31, 52, 53]’. 

53. Melzer A, Baudry C, Kadiri M, Ellis W. Tree use, feeding activity and diet of koalas on St Bees Island, Queensland. Australian Zoologist. 2011;35(3):870-5.

12. Line 320: The similar estimation of density resulting from RPAS and Spotlighting should be noted here. Although the effort is dissimilar, access to equipment may preclude the use of RPAS, so the information in Table 1 is useful and could be addressed here.

Author Response: We thank the reviewer for pointing out this opportunity. We have reworked the paragraph to place emphasis on the similarity in density estimates between RPAS, Spotlight and the naïve density, and the implications of using Spotlight when an RPAS in not available. See Paragraph L668-676.

13. Line 348: your data suggest that the SAT is not a good choice for this task, despite its appropriateness for detecting occupancy, so you could be firmer.

Author Response: We thank the reviewer for this suggestion. We have added a firmer statement at the end of the paragraph. See L707-709: ‘Given that Spotlight was a more reliable predictor of density compared to the SAT for sites in Port Stephens and Gilead, we conclude that practitioners should refrain from using the SAT to estimate koala density’.

Conclusions:

14. Line 355: The reference to winter is perhaps important? This is the first mention of it, but I presume this has to do with thermal target image and contrast. If so, comments regarding this are more appropriate than (for example) comments regarding wind speed, in my opinion. If this is a key limitation of the technique, please provide some background.

Author Response: Seasonal information for RPAS technology and koalas has been added to the introduction, methods and discussion. See L123-124, L134-141, 184-186, L425, L503, 

15. Line 361: I am unsure, given your results, why you would conclude that the method you are using is not a far superior approach, particularly at the landscape level where cost – effectiveness is vital. The required improvements you list to do not appear to have limited your study, so I find this confusing. This is particularly the case with comments regarding imagery. As a result, your conclusions sow a seed of doubt in the reader that (in my case anyway) did not previously exist.

Author Response: We agree and have removed the statement, as it creates confusion regarding the technology that in our view is not required for this manuscript.

In addition to the reviewers suggestions we have also made some minor edits to improve flow, and have added one additional reference that is relevant to our introduction (ref: 7). 

7. Crowther MS, Dargan JR, Madani G, Rus AI, Krockenberger MB, McArthur C, et al. Comparison of three methods of estimating the population size of an arboreal mammal in a fragmented rural landscape. Wildlife Research. 2020.

We trust that the response detailed above will be well-received and have addressed entirely the minor revisions required for our manuscript. 

With thanks,

Dr Ryan Witt | Conjoint Lecturer

School of Environmental and Life Sciences | Fauna Research Alliance Member 

Biology Building, The University of Newcastle (UON), University Drive, Callaghan NSW 2308 Australia

M: +61 (0)421 606 222 | E: ryan.witt@newcastle.edu.au

---

## [Editor Report · Decision Letter 1]

29 Oct 2020

Real-time drone derived thermal imagery outperforms traditional survey methods for an arboreal forest mammal

PONE-D-20-28042R1

Dear Dr. Witt,

We’re pleased to inform you that your manuscript has been judged scientifically suitable for publication and will be formally accepted for publication once it meets all outstanding technical requirements.

Kind regards,

Mathew S. Crowther, Ph.D

Academic Editor

PLOS ONE
---

## [Editor Report · Acceptance letter]

5 Nov 2020

PONE-D-20-28042R1 

Real-time drone derived thermal imagery outperforms traditional survey methods for an arboreal forest mammal 

Dear Dr. Witt:

I'm pleased to inform you that your manuscript has been deemed suitable for publication in PLOS ONE. Congratulations! Your manuscript is now with our production department. 

Kind regards, 

on behalf of

Assoc. Prof. Mathew S. Crowther 

Academic Editor

PLOS ONE